# T-Cell Responses Induced by an Intradermal BNT162b2 mRNA Vaccine Booster Following Primary Vaccination with Inactivated SARS-CoV-2 Vaccine

**DOI:** 10.3390/vaccines10091494

**Published:** 2022-09-07

**Authors:** Ratchanon Sophonmanee, Jomkwan Ongarj, Bunya Seeyankem, Purilap Seepathomnarong, Porntip Intapiboon, Smonrapat Surasombatpattana, Supattra Uppanisakorn, Pasuree Sangsupawanich, Sarunyou Chusri, Nawamin Pinpathomrat

**Affiliations:** 1Department of Biomedical Sciences and Biomedical Engineering, Faculty of Medicine, Prince of Songkla University, Songkhla 90110, Thailand; 2Department of Internal Medicine, Faculty of Medicine, Prince of Songkla University, Songkhla 90110, Thailand; 3Department of Pathology, Faculty of Medicine, Prince of Songkla University, Songkhla 90110, Thailand; 4Clinical Research Center, Faculty of Medicine, Prince of Songkla University, Songkhla 90110, Thailand

**Keywords:** intradermal, mRNA vaccine, inactivated SARS-CoV-2, COVID-19, T-cells

## Abstract

A practical booster vaccine is urgently needed to control the coronavirus disease (COVID-19) pandemic. We have previously reported the safety and immunogenicity of a fractional intradermal booster, using the BNT162b2 mRNA vaccine in healthy volunteers who had completed two doses of inactivated SARS-CoV-2 vaccine. In this study, an intramuscular booster at full dosage was used as a control, and a half-dose vaccination was included for reciprocal comparison. Detailed T-cell studies are essential to understand cellular responses to vaccination. T-cell immunity was examined using S1 peptide restimulation and flow cytometry. The fractional dose (1:5) of the BNT162b2 mRNA vaccine enhanced antigen-specific effector T-cells, but the responses were less remarkable compared to the intramuscular booster at full dosage. However, the intradermal regimen was not inferior to the intramuscular booster a month after boosting. An intradermal booster using only one-fifth of the standard dosage could provide comparable T-cell responses with the fractional intramuscular booster. This work confirms the efficacy of intradermal and fractional vaccination in terms of T-cell immunogenicity in previously immunised populations.

## 1. Introduction

In December 2019, a novel coronavirus (nCoV), the severe acute respiratory syndrome coronavirus 2 (SARS-CoV-2), was first detected in China. Two years since then, the world’s population has been facing the coronavirus disease 19 (COVID-19) pandemic [1]. On 13 January 2020, Thailand’s Ministry of Public Health (MoPH) reported an imported case of COVID-19, which was confirmed as the first laboratory-based 2019-nCoV from Wuhan, Hubei Province, China [2]. Globally, as of 6 June 2022, more than half a billion COVID-19 cases were confirmed, causing six million deaths [3]. The global economy has been affected severely by the COVID-19 pandemic. Because it is crucial to restrict the spread of the infection, most nations have opted for strict national lockdowns [4], severely affecting economic trade with different countries, leading to an economic crisis [5].

Achieving global COVID-19 vaccination targets as soon as possible will raise population immunity and significantly reduce the risk posed by new variants. It is necessary to continue monitoring novel variants, track vaccine effectiveness, and conduct research on the effects of additional doses, dosing schedules, and new vaccine formulations [6,7,8,9]. Immunogenicity studies in clinical trials have demonstrated that inactivated SARS-CoV-2 (CoronaVac) was immunogenic in adults aged 18–59 years and in older adults aged ≥60 years. The initial indication is that titres decline by 3 months after two doses [10,11]. Vaccination with a third-dose booster enhanced higher anti-SARS-CoV-2 receptor-binding domain (RBD) antibody titres compared to titres observed after primary vaccination in health professionals [12]. One month after the third dose, the authors found a significant increase in anti-RBD antibodies compared with baseline levels before the booster dose [13].

In a previous study, the immunisation and safety effects of the intradermal BNT162b2 mRNA vaccine booster after two doses of SARS-CoV-2 vaccination in healthy populations were satisfactory, although immune responses obtained from the reciprocal boosting were fewer than those obtained from the full-dose conventional booster. More importantly, there were fewer side effects after intradermal boosting [14]. Therefore, this study focuses on T-cell responses in healthy populations that have been administered two doses of inactivated SARS-CoV-2 (CoronaVac) and boosted with the intradermal BNT162b2 mRNA vaccine. The T-cell phenotype and effector cytokine production of S1-specific CD4^+^ and CD8^+^ T-cells were characterised. The results of this study may be used to inform clinical decisions and policymakers for the most efficient distribution of vaccines.

## 2. Materials and Methods

### 2.1. Study Procedures

This study was registered in the Thai Clinical Trials Registry (TCTR20211004001) and approved by the Human Research Ethics Committee (REC. 64-368-4-1). The trial was performed in accordance with the principles of Good Clinical Practice. Safety and theimmunogenicity data have been reported previously [14].

This study was conducted at the Clinical Research Center, Faculty of Medicine, Prince of Songkla University, Songkhla, Thailand. The demographics and CONSORT diagram of the study participants were shown in our previous study [14]. In brief, healthy volunteers aged 18–60 years (n = 91), having completed two doses of inactivated SARS-CoV-2 vaccine for 8–12 weeks were recruited and randomised to receive either 0.3 mL of BNT162b2 intramuscularly (n = 30), or a half dose (0.15 mL) of the mRNA vaccine intramuscularly (n = 30). In the intradermal group, 31 participants received one-fifth of the vaccine intradermally (n = 31). Blood samples were collected before and 14 and 28 days after the booster for T-cell analysis.

### 2.2. Sample Processing

Blood samples were collected in heparinised tubes and processed on the day of vaccination and then at 7, 14, and 28 days after the booster dose. To obtain peripheral blood mononuclear cells (PBMCs), the samples were centrifuged to separate the blood plasma, and the plasma was maintained at −80 °C. The remaining blood samples were diluted with RPMI (Gibco, Waltham, MA, USA) and placed into SepMATE tubes containing Lymphoprep (STEMCELL Technologies, Vancouver, Canada), according to the manufacturer’s protocol. The cell pellet was resuspended in 3 mL of R10 media (RPMI-1640 supplemented with 1% penicillin–streptomycin, 2 mM L-glutamine, and 10% foetal calf serum (FCS, Labtech, Heathfield, UK). The cells were adjusted to a concentration of 3 × 10^6^ PBMCs per mL before cryopreservation in FCS containing 10% DMSO. The cell suspensions were aliquoted and stored in CoolCell (Corning, Glendale, NY, USA) for freezing at −80 °C overnight. The tubes were stored in liquid nitrogen until further analysis.

### 2.3. Flow Cytometry Analysis

Flow cytometry analysis was performed on cryopreserved PBMCs, as previously described [15]. Briefly, cells were washed with R10, and each sample was stimulated with the S1 peptide pool (ProImmune, Oxford, UK), synthesised as 15-mers overlapping by ten amino acids (Appendix A) supplemented with anti-CD28 and CD49d. The working concentration of the peptides was 2 μg/mL. Cells were incubated for 18 h at 37 °C, 5% CO_2_, and GolgiPlug (Franklin, NJ, USA) was added after 2 h. Live/Dead Aqua (Invitrogen, Waltham, MA, USA) was diluted and used to stain the cells for 10 min, followed by surface staining with anti-CD3, CD4, and CD8 (BD, Franklin, NJ, USA) diluted in 2% Bovine Serum albumin (BSA) (Sigma-Aldrich, Saint Louis, MI, USA) in PBS for 30 min at 4 °C. The cells were then fixed and permeabilised using CytoFix (BD Biosciences, Franklin, NJ, USA) according to the manufacturer’s protocol. Cells were intracellularly stained with anti-IFN-γ, TNF-α (BD, Franklin, NJ, USA) diluted CytoPerm buffer (BD Biosciences, Franklin, NJ, USA) for 30 min at 4 °C. Antibody cocktails are shown in Appendix A. The cells were acquired using a CytoflexS (Beckman, Duarte, IN, USA). The data were analysed using FlowJo version 10 (FlowJo, Ashland, OR, USA) and gated, as shown in Appendix A.

### 2.4. Statistical Analysis

Statistical analyses were performed using GraphPad Prism 9 (GraphPad Software Inc., San Diego, CA, USA). To compare multiple groups, the Kruskal–Wallis test followed by Dunn’s multiple comparison test was performed. Differences were considered to be statistically significant at *p* ≤ 0.05.

## 3. Results

### 3.1. Effector CD4^+^ T-Cell Responses after the Booster

CD4^+^ T-cells play a role in enhancing specific immunity after vaccination. Blood samples were collected from volunteers and processed to obtain PBMCs. After restimulation with SARS-CoV-2 S1 peptides, the cells were stained for T-cell phenotypes and analysed using flow cytometry (Figure 1a). Eight to twelve weeks after receiving a two-dose regimen of inactivated SARS-CoV-2 vaccine, CD4^+^ T-cell responses were comparable in all study groups (Figure 1b). After an intradermal booster, the percentage of CD4^+^ T-cells increased from the baseline. However, the responses at Days 14 and 28 were not statistically different (Figure 1b). Effector cytokines were also evaluated by intracellular staining (Figure 1c,f). S1-specific IFN-γ producing CD4^+^ T-cells were higher at baseline in the full-dose IM group than in the half-dose IM group (Figure 1d) (*p* = 0.0287). At Day 14 after boosting, S1-specific IFN-γ-producing CD4^+^ T-cells showed a two-fold increase compared to baseline after the full dose intramuscular booster (IM full dose). Comparing the regimens, the responses remained significantly higher compared to the half-dose IM vaccination (*p* = 0.0491) and intradermal fractional vaccination (*p* = 0.0057) (Figure 1d). At Day 28, the IFN-γ-secreting CD4^+^ T-cell responses were waning in all groups and were not statistically different between regimens (Figure 1d). The TNF-α secretion was measured intracellularly (Figure 1f). The baseline responses of the TNF-α-secreting CD4^+^ T-cells were comparable between vaccination groups (Figure 1g). At Day 14 after boosting, TNF-α-producing CD4+ T-cells were inequitable between vaccination regimens in all study groups. Four weeks after the booster dose, cytokine production was increased in the IM half-dose group, and the responses remained significantly increased compared to those in the IM full-dose vaccination group (Figure 1f) (*p* = 0.0025). An intradermal booster using one-fifth of the standard dose failed to enhance the TNF-α response (Figure 1f).

In summary, IFN-γ-secreting CD4^+^ T-cells increased after being boosted with the mRNA vaccine. The full-dosage booster via the intramuscular route provided superior responses compared to fractional dosages intramuscularly or intradermally. The differences were not significant after four weeks of booster vaccination.

### 3.2. Effector CD8^+^ T-Cell Response after the Booster

CD8^+^ T-cells were analysed in this study to observe cytotoxic T-cell phenotypes and functions (Figure 2a). CD8^+^ T-cell responses at baseline were comparable between the vaccination regimens (Figure 2b). None of the booster regimens altered the percentage of CD8^+^ response (Figure 2b). Effector cytokine-producing CD8^+^ T-cells were identified by S1 restimulation and intracellular IFN-γ and TNF-α staining (Figure 2c,e). After 8–12 weeks of completed inactivated SARS-CoV-2 vaccination, the baseline responses of the effector cytokines were comparable between the vaccination groups. Two weeks after boosting, S1-specific IFN-γ-producing CD8^+^ T-cells in the IM full dose regimen increased and the responses remained significantly elevated compared to the fractional vaccination groups (IM half dose, *p* = 0.0182) (ID 1:5, *p* = 0.0318) (Figure 2d). The differences were no longer significant 4 weeks after the booster. However, the IFN-γ^+^ CD8^+^ T-cell response was sustained (Figure 2d). In contrast, none of the booster regimens improved the antigen-specific TNF-α responses of CD8^+^ T-cells at either time point (Figure 2f).

In conclusion, intramuscular booster using a standard dose enhanced IFN-γ-producing CD8^+^ T-cell responses compared with other boosting regimens. To reiterate, the responses were comparable after a month of boosting.

## 4. Discussion

This study focused on the differences in T-cell responses between vaccination regimens with a standard intramuscular booster and reciprocal intradermal booster using an mRNA vaccine (BNT162b2) in healthy adults who had completed a primary series of inactivated SARS-CoV-2 vaccines (CoronaVac). We observed significantly higher effector T-cell responses in the conventional booster than in the fractional vaccination dose via both injection routes. The intradermal vaccination showed non-inferior responses one month after follow-up compared to the standard booster.

Intradermal immunisation has been shown to reduce adverse reactions and enhance optimal immune responses using fewer vaccine doses [14,15,16]. Intapiboon et al. have shown that a fractional dose using one-fifth of the BNT162b2 mRNA vaccine provided less systemic side effects but could still enhance antibody responses, which provides an opportunity for dose escalation [14]. Consistent with the antibody data, cytokine-producing T-cell responses were enhanced after an intradermal booster using 1:5 BNT162b2, but the responses were less remarkable compared with those using the standard dose and route boosting. Conventional intramuscular vaccination using half of the vaccine dosage provided similar T-cell responses compared with 1:5 dose administered intradermally, agreeing with the IFN-γ-specific T-cell responses previously reported using enzyme-linked immune absorbent spot (ELISpot) [14]. The results from this study confirm that intradermal mRNA vaccination can improve T-cell responses from the previous inactivated SARS-CoV-2 vaccination in a manner similar to that of fractional intramuscular boosters.

After 8 weeks of the completed inactivated SARS-CoV-2 regimen, T-cell responses were diminished, requiring a booster dose [14,15,17]. The S1-specific T-cell immunity of the participants was low in both CD4^+^ and CD8^+^ T-cell responses. Without a booster dose, these participants were at risk of SARS-CoV-2 infection and severe clinical outcomes [18]. The recent outbreak of Omicron strains could be explained by waning immunity after an inadequate vaccination program and lack of booster vaccination [8,19]. Our findings provide an alternative vaccination strategy for increasing vaccine coverage and boosting host cellular immunity against emerging viral strains.

T-cell studies are required to observe vaccine immunogenicity in preclinical and early clinical studies [20,21]. Immunological methods for T-cell analysis vary in each study. IFN-γ-producing T-cells are measured using ELISpot to observe total cellular responses; however, flow cytometry is preferred when analysing detailed T-cell phenotypes and cytokines [21,22]. In this study, we observed a higher response of IFN-γ-producing T-cells after boosting with the intramuscular mRNA vaccine booster than after intradermal injection. This is consistent with previous studies on T-cell responses after vaccination with two doses of intramuscular BNT162b2 [23]. Only one study has reported T-cell responses after intradermal vaccination with BNT162b2, showing similar responses between vaccination routes [14]. Intradermal vaccination consistently provides good T-cell responses that are not limited to mRNA vaccines. Comparable T-cell responses between intradermal and intramuscular vaccinations were detected after immunisation with a ChAd63 viral vector vaccine expressing malarial antigens as well as ChAdOx1 nCoV-19 [15,16]. A fractional dose of ChAdOx1 nCoV-19 administered intradermally could enhance antigen-specific T-cell responses that produce effector cytokines [15]. T-cell immunity has been shown to play a prominent role in controlling disease severity [24]. Interestingly, antigen-specific T-cells that were induced by infection or vaccination with parental strains could cross-react with new variant peptides [25,26]. This suggests that the obtained T-cell responses should be able to counter variants of concern (VOCs). However, no study has reported T-cell functions induced by intradermal vaccination against VOCs.

A few recent studies on fractional and intradermal vaccinations have been published [14,15,17,27,28]. The immunogenicity of humoral and cellular responses obtained from a booster dose depends on various factors, such as vaccine type, dosage, and route of vaccination. We previously reported that intradermal vaccination using ChAdOx1 nCoV19 is non-inferior to other routes [15]. However, this was not the case for the BNT162b2 mRNA booster, which seemed to be dose-dependent [14]. Administering one-fifth of the dose ID or half-dose IM of the mRNA vaccine would result in fewer antibody responses [14]. However, intradermal vaccination still has a great benefit. Many low- and middle-income countries still have limited access to COVID-19 vaccines; therefore, fractional vaccination could be a reasonable option for increasing vaccine coverage [29,30]. Intradermal BCG vaccination is routinely performed in newborns in these countries [31,32] which means that healthcare professionals can deliver intradermal injections for COVID-19 vaccines as well.

This study had several limitations. Due to the duration of the study, the association between T-cell immunogenicity, infection, and disease severity was not analysed. This study focused on volunteers previously vaccinated with inactivated SARS-CoV-2. However, the application of these findings in other vaccinated populations is limited. The BNT162b2 booster was examined in this study and cannot be directly compared with other vaccine types. This study included a duration of more than 8 weeks after completion of the CoronaVac vaccination and booster doses. Thus, its application may be limited to shorter intervals. Finally, because the quantity of collected PBMCs was limited, other T- or B-cell phenotypes were not analysed.

Further evaluation of memory T-cell immunity after intradermal booster of the BNT162b2 vaccine is important to observe the longevity of antigen-specific T-cell repertoires.

## 5. Conclusions

The phase I clinical trial of an intradermal BNT162b2 mRNA booster after the completion of a primary series of inactivated SARS-CoV-2 vaccines was previously reported by our group. Serological responses were shown together with simple cellular immunity analysis. Here, we provide more details on effector T-cell responses, including T-cell phenotypes and intracellular cytokine production. Consistent with the previous report, the T-cell responses obtained from the fractional booster were improved, but the levels were lower than those of the conventional booster. Cytokine-producing T-cell responses were greatly enhanced within 2 weeks, especially when boosted with the conventional dose and route. However, the responses between all regimens were comparable after a month, making fractional intradermal vaccination non-inferior to intramuscular booster using higher vaccine dosages. Our results confirm that intradermal booster using only one-fifth of the standard dosage is a non-inferior regimen. This rational booster can be applied to a population previously immunised with inactivated vaccines.

## Figures and Tables

**Figure 1 vaccines-10-01494-f001:**
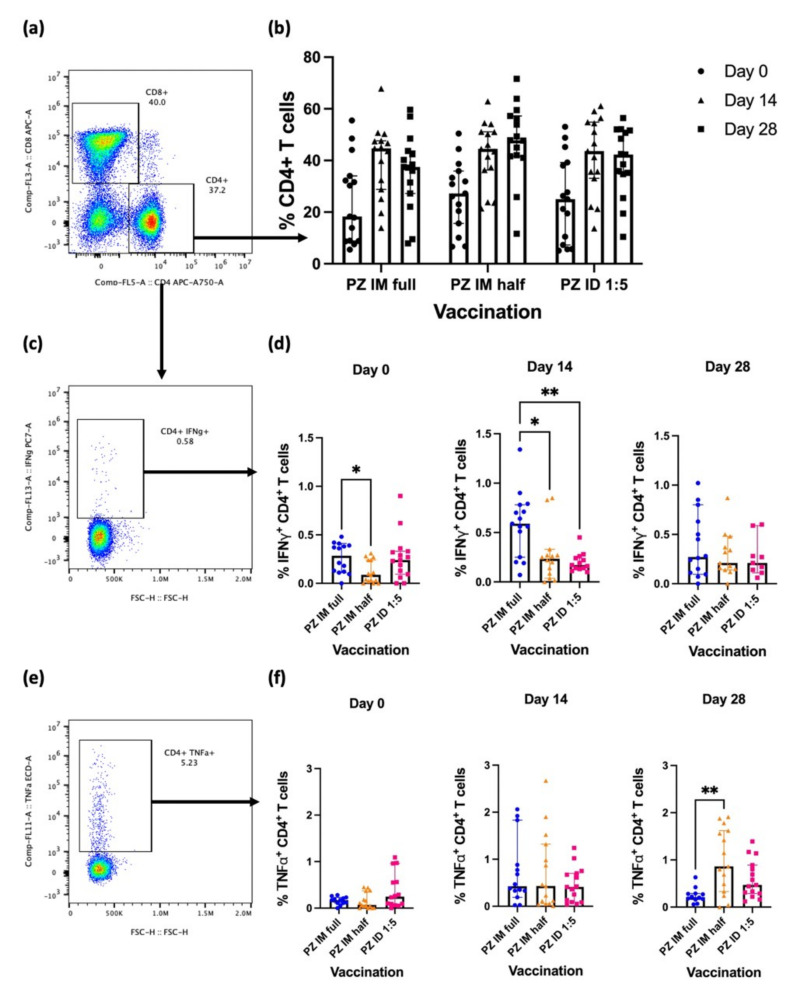
Effector cytokine production of S1-specific CD4^+^ T-cells after BNT162b2 boosting. All volunteers were previously vaccinated with two doses of inactivated SARS-CoV-2 vaccines (n = 91). The BNT162b2 mRNA booster was administered in three groups: full dose intramuscularly (n = 30), half dose intramuscularly (n = 30), and one-fifth vaccine dose intradermally (n = 31). (**a**) CD4^+^ and CD8^+^ T-cell populations, (**b**) percentage of CD4^+^ T-cells at Day 0 (before booster dose), Day 14, and Day 28 (after booster dose), (**c**) S1-specific IFN-γ-producing CD4^+^ T-cells, (**d**) percentage of S1-specific IFN-γ-producing CD4^+^ T-cell responses at Day 0 (before booster dose), Day 14, and Day 28 (after booster dose), (**e**) S1-specific TNF-α-producing CD4^+^ T-cells, (**f**) Percentage of S1-specific TNF-α producing CD4^+^ T-cell responses at Day 0 (before booster dose), Day 14, and Day 28 (after booster dose). Each symbol represents the median of one participant with 95% CI (n = 30–31 volunteers). Statistical significance was determined using the Kruskal–Wallis’s test, with Dunn’s multiple comparison test between vaccinated groups. * *p* ≤ 0.05; ** *p* ≤ 0.01.

**Figure 2 vaccines-10-01494-f002:**
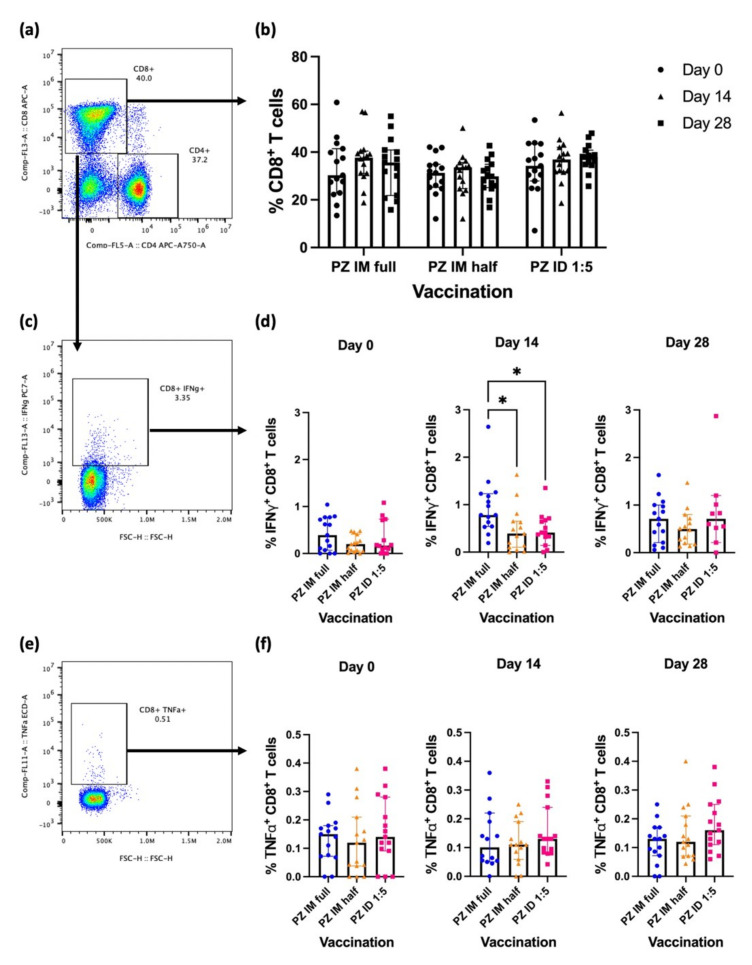
Effector cytokine production of S1-specific CD8^+^ T-cells after BNT162b2 boosting. All volunteers were previously vaccinated with two doses of inactivated SARS-CoV-2 vaccines (n = 91). The BNT162b2 mRNA booster vaccine was administered in three groups: intramuscularly with either full dosage (n = 30), or half dosage (n = 30), and intradermally with fractional dosage (n = 31). (**a**) CD4^+^ T-cell and CD8^+^ T-cell populations, (**b**) percentage of CD8^+^ T-cells at Day 0 (before booster dose), Day 14 and Day 28 (after booster dose), (**c**) S1-specific IFN-γ-producing CD8^+^ T-cells, (**d**) percentage of S1-specific IFN-γ-producing CD8^+^ T-cell responses at Day 0 (before booster dose), Day 14, and Day 28 (after booster dose), (**e**) S1-specific TNF-α-producing CD8^+^ T-cells, (**f**) percentage of S1-specific TNF-α-producing CD8^+^ T-cell responses at Day 0 (before booster dose), Day 14, and Day 28 (after booster dose). Each symbol represents the median of one participant with 95% CI (n = 30–31 volunteers). Statistical significance was determined using the Kruskal–Wallis’s test, with Dunn’s multiple comparison test between vaccinated groups. * *p* ≤ 0.05.

## Data Availability

The data supporting the findings of this study are available from the corresponding author upon reasonable request.

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
