# Peer review of "T-Cell Responses Induced by an Intradermal BNT162b2 mRNA Vaccine Booster Following Primary Vaccination with Inactivated SARS-CoV-2 Vaccine"

_vaccines, 2022, doi:10.3390/vaccines10091494_

Round 1
Reviewer 1 Report
In this short report, Sophonmanee et al. builds up on a previous report by the same group, where they describe the immune responses to a intradermal immunization with the BNT162b2 SARS-CoV2 mRNA vaccine, in individuals previously immunized with inactivated SARS-CoV2 vaccines (CoronaVac). This is a phase I clinical trial. In the results, the authors show that CD4 and CD8 T cell responses are promoted somewhat comparably to individuals immunized intramuscularly with a third dose of CoronaVac.
Overall, the results are well-presented, conclusions are supported by data (and not overinterpreted), and are important for our understanding on how the human immune system responds to distinct routes of immunization against SARS-CoV2 - including routes and immunizations with increased safety. This article needs, however, some minor adjustments that would further increase the quality of this report:
1 - Language editing would be desirable to improve the quality of scientific communication.
2 - Please add a conclusion statement following each results section.
3 - If possible, a measurement of Gzm-B production by CD8+ T cells would also be advantageous.
4 - In addition, an evaluation of memory vs effector markers (e.g. CCR7, CD45RA, CD45RO) would also add to our understanding of how the ID mRNA vaccine adds to the immune response to SARS-CoV2.
5 - Figures 1/2: Please alter the yellow color coding, to improve readability.
Author Response
Reviewer 1:
Comments and Suggestions for Authors
In this short report, Sophonmanee et al. builds up on a previous report by the same group, where they describe the immune responses to a intradermal immunization with the BNT162b2 SARS-CoV2 mRNA vaccine, in individuals previously immunized with inactivated SARS-CoV2 vaccines (CoronaVac). This is a phase I clinical trial. In the results, the authors show that CD4 and CD8 T cell responses are promoted somewhat comparably to individuals immunized intramuscularly with a third dose of CoronaVac.
Overall, the results are well-presented, conclusions are supported by data (and not overinterpreted), and are important for our understanding on how the human immune system responds to distinct routes of immunization against SARS-CoV2 - including routes and immunizations with increased safety. This article needs, however, some minor adjustments that would further increase the quality of this report:
Point 1: Language editing would be desirable to improve the quality of scientific communication.
Response 1: We thank the reviewer for the suggestion. English language editing has now been done as suggested.
Point 2: Please add a conclusion statement following each results section.
Response 2: We thank the reviewer for the suggestion. The conclusion is now included statement following each results section as suggested.
Point 3: If possible, a measurement of Gzm-B production by CD8+ T cells would also be advantageous.
Response 3: We thank the reviewer for the valuable comment. Granzyme B is indeed very interesting to measure cytotoxic activity of the CD8+ T cells. However, in this study we aim to examine mainly Th1 responses using IFNg production. Effector CD8+ T cells could also be identified using the IFNg secretion. We have limited flow cytometry lasers and fluorochromes. Therefore, we have only included IFNg and TNFa for the intracellular cytokine examination in our current flow panel.
Point 4: In addition, an evaluation of memory vs effector markers (e.g. CCR7, CD45RA, CD45RO) would also add to our understanding of how the ID mRNA vaccine adds to the immune response to SARS-CoV2.
Response 4: We thank the reviewer for the valuable comment. We totally agree that it would be very interesting to explore the different memory T cell phenotypes between the vaccine route (ID vs IM). In this study we focused only on the effector T cell responses as the samples were obtained before and short after booster (Day 14 and D24). We are now in the process of analysing the PBMC samples from Day 90 post booster to observe memory vs effector responses using CCR7 CD45RO markers. We hope to see the differences between the route of vaccination and will report this data set in the next article.
Point 5: Figures 1/2: Please alter the yellow color coding, to improve readability.
Response 5: We thank the reviewer for the helpful suggestion. We have now changed the yellow coding to a darker colour for better readability.
Reviewer 2 Report
This is a continual report of author Nawamin’s previous publication “Immunogenicity and safety of an intradermal BNT162b2 mRNA vaccine booster after two doses of inactivated SARS-CoV-2 vaccine in healthy population (reference 9).”
In the previous report, authors had shown that intramuscular administration of BNT162b2 mRNA booster vaccine could evidently increase the number of IFNg secreting T cells, in either full- or half dosage intramuscular injection group. Intradermal administration of BNT162b2 mRNA booster shot, however, did not clearly elevate the number of IFNg secreting T cells.
Number of IFNg secreting T cells was about 50-80 per million of PBMCs in healthy individuals who had two conventional doses of inactivated SARS-CoV-2 vaccine (Figure 5 in reference 9).
In this study, investigators further used fluorescent surface marker staining, flow cytometry and statistical analysis of flow data to categorize subtypes of IFNg secreting T cells and to compare the difference among the three dosage/injection site groups.
This study would be more valuable, if more rationality and scrupulousness were properly arranged in data presentation and the writing.
Major Suggestions:
1. Language: (A) The title showed some scientific merit of a clinical study. However, context of the manuscript contained a number of misused, redundant writing as well as grammatical errors. The manuscript needs help in English writing. (B) Moreover, following precisely the consensus usage of words would make more sense. Although the authors could argue for the academic freedom in this specific manuscript, it is a fact that it would be technically hard to apply intradermal injection for population vaccination in humans. Some authors from other research field and published papers already noticed such contradictions (Please check the other published papers on the web, intradermal vaccination is for animals, and subcutaneous injection is for humans). Interestingly, Michael Ryan in WHO had pointed out that herd immunity is for stock animals, and population vaccination is for human beings.
2. Concept and data presentation:
(1) To avoid questionable concept and more precise writing is required: (A) Please tell the truth of the study and write accordingly. The study population had been recruited at least two years ago, and in relation to the different dosage/injection site divided into three categories: intramuscular administration of BNT162b2 mRNA booster vaccine with either full- or half dosage, and the third group was those who had intradermal injection group with fractional vaccine. Therefore, the standard control cannot be written as “An intramuscular booster with the full dosage was recruited as a standard control ---.” Though, it could be stated as: the intramuscular booster with the full dosage was used as a control in this study. (B) In abstract, it was stated that details of T cell study are essential to support the serology data. Nonetheless, no serology data were presented anywhere in this manuscript.
(2) Requires adequate data presentation, not just stating the results without data. Please check the last four sentences at the end of the abstract and the lack of results.
Author Response
Reviewer 2:
Comments and Suggestions for Authors
This is a continual report of author Nawamin’s previous publication “Immunogenicity and safety of an intradermal BNT162b2 mRNA vaccine booster after two doses of inactivated SARS-CoV-2 vaccine in healthy population (reference 9).”
In the previous report, authors had shown that intramuscular administration of BNT162b2 mRNA booster vaccine could evidently increase the number of IFNg secreting T cells, in either full- or half dosage intramuscular injection group. Intradermal administration of BNT162b2 mRNA booster shot, however, did not clearly elevate the number of IFNg secreting T cells.
Number of IFNg secreting T cells was about 50-80 per million of PBMCs in healthy individuals who had two conventional doses of inactivated SARS-CoV-2 vaccine (Figure 5 in reference 9).
In this study, investigators further used fluorescent surface marker staining, flow cytometry and statistical analysis of flow data to categorize subtypes of IFNg secreting T cells and to compare the difference among the three dosage/injection site groups.
This study would be more valuable, if more rationality and scrupulousness were properly arranged in data presentation and the writing.
Major Suggestions:
Language:
Point 1: The title showed some scientific merit of a clinical study. However, context of the manuscript contained a number of misused, redundant writing as well as grammatical errors. The manuscript needs help in English writing.
Response 1: We thank the reviewer for the helpful suggestion. English language editing has now been used to help for clearer data description as suggested.
Point 2: Moreover, following precisely the consensus usage of words would make more sense. Although the authors could argue for the academic freedom in this specific manuscript, it is a fact that it would be technically hard to apply intradermal injection for population vaccination in humans. Some authors from other research field and published papers already noticed such contradictions (Please check the other published papers on the web, intradermal vaccination is for animals, and subcutaneous injection is for humans). Interestingly, Michael Ryan in WHO had pointed out that herd immunity is for stock animals, and population vaccination is for human beings.
Response 2: We thank the reviewer for the helpful comment. We aware of the recent publications in the fields of fractional and intradermal vaccination. The immunogenicity of the humoral and cellular responses obtained from a booster dose depends on various factors such as vaccine types, dosages as well as routes of vaccination. We have reported the non-inferior study of intradermal vaccination using ChAdOx1 nCoV19. However, it was not the case for the BNT162b2 mRNA booster which seems to be dose dependent. Giving the vaccine one fifth of the dose ID or half dose IM would obtain less antibody responses. However, we still believe that there is still a great benefit in the intradermal vaccination. Many developing countries are still had limited access to the COVID-19 vaccines so fractional dose would help with the vaccine coverage. Intradermal injection is one of skills that healthcare professionals in LMIC can perform. Intradermal BCG vaccination is routinely performed in human newborns in these countries including Thailand where all the nurses and doctors can do intradermal injection. Although, this report focuses on T cell immunogenicity not the technical issue of intradermal vaccination.
Concept and data presentation:
Point 3: To avoid questionable concept and more precise writing is required: (A) Please tell the truth of the study and write accordingly. The study population had been recruited at least two years ago, and in relation to the different dosage/injection site divided into three categories: intramuscular administration of BNT162b2 mRNA booster vaccine with either full- or half dosage, and the third group was those who had intradermal injection group with fractional vaccine. Therefore, the standard control cannot be written as “An intramuscular booster with the full dosage was recruited as a standard control ---.” Though, it could be stated as: the intramuscular booster with the full dosage was used as a control in this study.
Response 3: We thank the reviewer for the helpful comment. We totally agree with the reviewer and have revised the control statement as suggested.
Point 4: (B) In abstract, it was stated that details of T cell study are essential to support the serology data. Nonetheless, no serology data were presented anywhere in this manuscript.
Response 4: We thank the reviewer for the helpful comment. The serology data were not indeed included in this report. They were published earlier in our previous report. We have revised the sentence in the abstract to precisely say what is shown in this current report.
Point 5: Requires adequate data presentation, not just stating the results without data. Please check the last four sentences at the end of the abstract and the lack of results.
Response 5: We thank the reviewer for the helpful comment. We wrote the abstract based on the data from this report that is continued from the previous safety and immunogenicity data. We have now revised the end of the abstract to clearly explain that is included in this report as suggested.
Reviewer 3 Report
In the article titled, “T cell responses induced by an intradermal BNT162b2 mRNA 2 vaccine booster followed primary vaccination of inactivated 3 SARS-CoV-2 vaccine”, Sophonmanee et.al compare CD4 and CD8 T cell responses in healthy controls who received 2 doses of inactivated SARS-CoV-2 (CoronaVac) vaccine and an intradermal booster shot of BNT162b2 mRNA vaccine by flowcytometry.
Strengths-
· This study adds to the literature the knowledge about T cell and cytokine responses after receiving 2 doses of vaccine and a booster shot.
Weakness/comments-
1. Materials and Methods
· What kind of blood tubes were used to collect whole blood?
2. Results
· English language and grammar can be improved by a native English speaker
Author Response
Reviewer 3:
Comments and Suggestions for Authors
In the article titled, “T cell responses induced by an intradermal BNT162b2 mRNA 2 vaccine booster followed primary vaccination of inactivated 3 SARS-CoV-2 vaccine”, Sophonmanee et.al compare CD4 and CD8 T cell responses in healthy controls who received 2 doses of inactivated SARS-CoV-2 (CoronaVac) vaccine and an intradermal booster shot of BNT162b2 mRNA vaccine by flowcytometry.
Strengths-
· This study adds to the literature the knowledge about T cell and cytokine responses after receiving 2 doses of vaccine and a booster shot.
Weakness/comments-
Point 1: Materials and Methods What kind of blood tubes were used to collect whole blood?
Response 1: We thank the reviewer for the comment. We used heparinised tube for collecting blood samples for T cell studies. We have now included this information in the Materials and Methods
Point 2: Results English language and grammar can be improved by a native English speaker
Response 2: We thank the reviewer for the helpful suggestion. English language editing has now been used to improve the English writing as suggested.
Round 2
Reviewer 2 Report
The manuscript is acceptable in present form.